# Autoencoding Variational Inference
# For Topic Models

**Akash Srivastava**
Informatics Forum, University of Edinburgh
10, Crichton St
Edinburgh, EH89AB, UK
`akash.srivastava@ed.ac.uk`

**Charles Sutton**[*]
Informatics Forum, University of Edinburgh
10, Crichton St
Edinburgh, EH89AB, UK
`csutton@inf.ed.ac.uk`

## Abstract

Topic models are one of the most popular methods for learning representations of text, but a major challenge is that any change to the topic model requires mathematically deriving a new inference algorithm. A promising approach to address this problem is autoencoding variational Bayes (AEVB), but it has proven difficult to apply to topic models in practice. We present what is to our knowledge the first effective AEVB based inference method for latent Dirichlet allocation (LDA), which we call Autoencoded Variational Inference For Topic Model (AVITM). This model tackles the problems caused for AEVB by the Dirichlet prior and by component collapsing. We find that AVITM matches traditional methods in accuracy with much better inference time. Indeed, because of the inference network, we find that it is unnecessary to pay the computational cost of running variational optimization on test data. Because AVITM is black box, it is readily applied to new topic models. As a dramatic illustration of this, we present a new topic model called ProdLDA, that replaces the mixture model in LDA with a product of experts. By changing only one line of code from LDA, we find that ProdLDA yields much more interpretable topics, even if LDA is trained via collapsed Gibbs sampling.

## 1 Introduction

Topic models (Blei, 2012) are among the most widely used models for learning unsupervised representations of text, with hundreds of different model variants in the literature, and have have found applications ranging from the exploration of the scientific literature (Blei & Lafferty, 2007) to computer vision (Fei-Fei & Perona, 2005), bioinformatics (Rogers et al., 2005), and archaeology (Mimno, 2009). A major challenge in applying topic models and developing new models is the computational cost of computing the posterior distribution. Therefore a large body of work has considered approximate inference methods, the most popular methods being variational methods, especially mean field methods, and Markov chain Monte Carlo, particularly methods based on collapsed Gibbs sampling.

Both mean-field and collapsed Gibbs have the drawback that applying them to new topic models, even if there is only a small change to the modeling assumptions, requires re-deriving the inference methods, which can be mathematically arduous and time consuming, and limits the ability of practitioners to freely explore the space of different modeling assumptions. This has motivated the development of black-box inference methods (Ranganath et al., 2014; Mnih & Gregor, 2014; Kucukelbir et al., 2016; Kingma & Welling, 2014) which require only very limited and easy to compute information from the model, and hence can be applied automatically to new models given a simple declarative specification of the generative process.

Autoencoding variational Bayes (AEVB) (Kingma & Welling, 2014; Rezende et al., 2014) is a particularly natural choice for topic models, because it trains an *inference network* (Dayan et al., 1995), a neural network that directly maps a document to an approximate posterior distribution,

---

[*]Additional affiliation: Alan Turing Institute, British Library, 96 Euston Road, London NW1 2DB

without the need to run further variational updates. This is intuitively appealing because in topic models, we expect the mapping from documents to posterior distributions to be well behaved, that is, that a small change in the document will produce only a small change in topics. This is exactly the type of mapping that a universal function approximator like a neural network should be good at representing. Essentially, the inference network learns to mimic the effect of probabilistic inference, so that on test data, we can enjoy the benefits of probabilistic modeling without paying a further cost for inference.

However, despite some notable successes for latent Gaussian models, black box inference methods are significantly more challenging to apply to topic models. For example, in initial experiments, we tried to apply ADVI (Kucukelbir et al., 2016), a recent black-box variational method, but it was difficult to obtain any meaningful topics. Two main challenges are: first, the Dirichlet prior is not a location scale family, which hinders reparameterisation, and second, the well known problem of component collapsing (Dinh & Dumoulin, 2016), in which the inference network becomes stuck in a bad local optimum in which all topics are identical.

In this paper, we present what is, to our knowledge, the first effective AEVB inference method for topic models, which we call Autoencoded Variational Inference for Topic Models or AVITM[1]. On several data sets, we find that AVITM yields topics of equivalent quality to standard mean-field inference, with a large decrease in training time. We also find that the inference network learns to mimic the process of approximate inference highly accurately, so that it is not necessary to run variational optimization at all on test data.

But perhaps more important is that AVITM is a black-box method that is easy to apply to new models. To illustrate this, we present a new topic model, called ProdLDA, in which the distribution over individual words is a product of experts rather than the mixture model used in LDA. We find that ProdLDA consistently produces better topics than standard LDA, whether measured by automatically determined topic coherence or qualitative examination. Furthermore, because we perform probabilistic inference using a neural network, we can fit a topic model on roughly a one million documents in under 80 minutes on a single GPU, and because we are using a black box inference method, implementing ProdLDA requires a change of *only one line of code* from our implementation of standard LDA.

To summarize, the main advantages of our methods are:

1. *Topic coherence:* ProdLDA returns consistently better topics than LDA, even when LDA is trained using Gibbs sampling.
2. *Computational efficiency:* Training AVITM is fast and efficient like standard mean-field. On new data, AVITM is much faster than standard mean field, because it requires only one forward pass through a neural network.
3. *Black box:* AVITM does not require rigorous mathematical derivations to handle changes in the model, and can be easily applied to a wide range of topic models.

Overall, our results suggest that AVITM is ready to take its place alongside mean field and collapsed Gibbs as one of the workhorse inference methods for topic models.

## 2 Background

To fix notation, we begin by describing topic modelling and AVITM.

### 2.1 Latent Dirichlet Allocation

We describe the most popular topic model, latent Dirichlet allocation (LDA). In LDA, each document of the collection is represented as a mixture of topics, where each topic $\beta_k$ is a probability distribution over the vocabulary. We also use $\beta$ to denote the matrix $\beta = (\beta_1 \ldots \beta_K)$. The generative process is then as described in Algorithm 1. Under this generative model, the marginal likelihood of

---

[1]Code available at
https://github.com/akashgit/autoencoding_vi_for_topic_models

**for** *each document* **w do**
 Draw topic distribution $\theta \sim$ Dirichlet($\alpha$);
 **for** *each word at position* $n$ **do**
 Sample topic $z_n \sim$ Multinomial($1, \theta$);
 Sample word $w_n \sim$ Multinomial($1, \beta_{z_n}$);
 **end**
**end**

**Algorithm 1:** LDA as a generative model.

a document **w** is

$$p(\mathbf{w}|\alpha, \beta) = \int_\theta \left( \prod_{n=1}^{N} \sum_{z_n=1}^{k} p(w_n|z_n, \beta) p(z_n|\theta) \right) p(\theta|\alpha) d\theta. \tag{1}$$

Posterior inference over the hidden variables $\theta$ and $z$ is intractable due to the coupling between the $\theta$ and $\beta$ under the multinomial assumption (Dickey, 1983).

## 2.2 MEAN FIELD AND AEVB

A popular approximation for efficient inference in topic models is mean field variational inference, which breaks the coupling between $\theta$ and $z$ by introducing free variational parameters $\gamma$ over $\theta$ and $\phi$ over $z$ and dropping the edges between them. This results in an approximate variational posterior $q(\theta, z|\gamma, \phi) = q_\gamma(\theta) \prod_n q_\phi(z_n)$, which is optimized to best approximate the true posterior $p(\theta, z|\mathbf{w}, \alpha, \beta)$. The optimization problem is to minimize

$$L(\gamma, \phi \mid \alpha, \beta) = D_{KL}\left[q(\theta, z|\gamma, \phi)||p(\theta, z|\mathbf{w}, \alpha, \beta)\right] - \log p(\mathbf{w}|\alpha, \beta). \tag{2}$$

In fact the above equation is a lower bound to the marginal log likelihood, sometimes called an *evidence lower bound (ELBO)*, a fact which can be easily verified by multiplying and dividing (1) by the variational posterior and then applying Jensen's inequality on its logarithm. Note that the mean field method optimizes over an independent set of variational parameters for each document. To emphasize this, we will refer to this standard method by the non-standard name of *Decoupled Mean-Field Variational Inference* (DMFVI).

For LDA, this optimization has closed form coordinate descent equations due to the conjugacy between the Dirichlet and multinomial distributions. Although this is a computationally convenient aspect of DMFVI, it also limits its flexibility. Applying DMFVI to new models relies on the practitioner's ability to derive the closed form updates, which can be impractical and sometimes impossible.

AEVB (Kingma & Welling, 2014; Rezende et al., 2014) is one of several recent methods that aims at "black box" inference methods to sidestep this issue. First, rewrite the ELBO as

$$L(\gamma, \phi \mid \alpha, \beta) = -D_{KL}\left[q(\theta, z|\gamma, \phi)||p(\theta, z|\alpha)\right] + \mathbb{E}_{q(\theta, z|\gamma, \phi)}[\log p(\mathbf{w}|z, \theta, \alpha, \beta)] \tag{3}$$

This form is intuitive. The first term attempts to match the variational posterior over latent variables to the prior on the latent variables, while the second term ensures that the variational posterior favors values of the latent variables that are good at explaining the data. By analogy to autoencoders, this second term is referred to as a *reconstruction term*.

What makes this method "Autoencoding," and in fact the main difference from DMFVI, is the parameterization of the variational distribution. In AEVB, the variational parameters are computed by using a neural network called an *inference network* that takes the observed data as input. For example, if the model prior $p(\theta)$ were Gaussian, we might define the inference network as a feedforward neural network $(\mu(\mathbf{w}), \mathbf{v}(\mathbf{w})) = f(\mathbf{w}, \gamma)$, where $\mu(\mathbf{w})$ and $\mathbf{v}(\mathbf{w})$ are both vectors of length $k$, and $\gamma$ are the network's parameters. Then we might choose a Gaussian variational distribution $q_\gamma(\theta) = N(\theta; \mu(\mathbf{w}), \text{diag}(\mathbf{v}(\mathbf{w})))$, where diag($\cdots$) produces a diagonal matrix from a column vector. The variational parameters $\gamma$ can then be chosen by optimizing the ELBO (3). Note that we have

now, unlike DMFVI, coupled the variational parameters for different documents because they are all computed from the same neural network. To compute the expectations with respect to $q$ in (3), Kingma & Welling (2014); Rezende et al. (2014) use a Monte Carlo estimator which they call the "reparameterization trick" (RT; appears also in Williams (1992)). In the RT, we define a variate $U$ with a simple distribution that is independent of all variational parameters, like a uniform or standard normal, and a reparameterization function $F$ such that $F(U, \gamma)$ has distribution $q_\gamma$. This is always possible, as we could choose $F$ to be the inverse cumulative distribution function of $q_\gamma$, although we will additionally want $F$ to be easy to compute and differentiable. If we can determine a suitable $F$, then we can approximate (3) by taking Monte Carlo samples of $U$, and optimize $\gamma$ using stochastic gradient descent.

# 3 AUTOENCODING VARIATIONAL BAYES IN LATENT DIRICHLET ALLOCATION

Although simple conceptually, applying AEVB to topic models raises several practical challenges. The first is the need to determine a reparameterization function for $q(\theta)$ and $q(z_n)$ to use the RT. The $z_n$ are easily dealt with, but $\theta$ is more difficult; if we choose $q(\theta)$ to be Dirichlet, it is difficult to apply the RT, whereas if we choose $q$ to be Gaussian or logistic normal, then the KL divergence in (3) becomes more problematic. The second issue is the well known problem of component collapsing (Dinh & Dumoulin, 2016), which a type of bad local optimum that is particularly endemic to AEVB and similar methods. We describe our solutions to each of those problems in the next few subsections.

## 3.1 COLLAPSING z'S

Dealing with discrete variables like **z** using reparameterization can be problematic, but fortunately in LDA the variable **z** can be conveniently summed out. By collapsing **z** we are left with having to sample from $\theta$ only, reducing (1) to

$$p(\mathbf{w}|\alpha, \beta) = \int_\theta \left( \prod_{n=1}^{N} p(w_n|\beta, \theta) \right) p(\theta|\alpha)d\theta. \tag{4}$$

where the distribution of $w_n|\beta, \theta$ is Multinomial(1, $\beta\theta$), recalling that $\beta$ denotes the matrix of all topic-word probability vectors.

## 3.2 WORKING WITH DIRICHLET BELIEFS: LAPLACE APPROXIMATION

LDA gets its name from the Dirichlet prior on the topic proportions $\theta$, and the choice of Dirichlet prior is important to obtaining interpretable topics (Wallach et al., 2009). But it is difficult to handle the Dirichlet within AEVB because it is difficult to develop an effective reparameterization function for the RT. Fortunately, a RT does exist for the Gaussian distribution and has been shown to perform quite well in the context of variational autoencoder (VAE) (Kingma & Welling, 2014).

We resolve this issue by constructing a Laplace approximation to the Dirichlet prior. Following MacKay (1998), we do so in the softmax basis instead of the simplex. There are two benefits of this choice. First, Dirichlet distributions are unimodal in the softmax basis with their modes coinciding with the means of the transformed densities. Second, the softmax basis also allows for carrying out unconstrained optimization of the cost function without the simplex constraints. The Dirichlet probability density function in this basis over the softmax variable **h** is given by

$$P(\theta(\mathbf{h})|\alpha) = \frac{\Gamma(\sum_k \alpha_k)}{\prod_k \Gamma(\alpha_k)} \prod_k \theta_k^{\alpha_k} g(\mathbf{1}^T \mathbf{h}). \tag{5}$$

Here $\theta = \sigma(\mathbf{h})$, where $\sigma(.)$ represents the softmax function. Recall that the Jacobian of $\sigma$ is proportional to $\prod_k \theta_k$ and $g(\cdot)$ is an arbitrary density that ensures integrability by constraining the redundant degree of freedom. We use the Laplace approximation of Hennig et al. (2012), which

has the property that the covariance matrix becomes diagonal for large $k$ (number of topics). This approximation to the Dirichlet prior $p(\theta|\alpha)$ is results in the distribution over the softmax variables $\mathbf{h}$ as a multivariate normal with mean $\mu_1$ and covariance matrix $\Sigma_1$ where

$$\mu_{1k} = \log \alpha_k - \frac{1}{K} \sum_i \log \alpha_i$$

$$\Sigma_{1kk} = \frac{1}{\alpha_k} \left( 1 - \frac{2}{K} \right) + \frac{1}{K^2} \sum_i \frac{1}{\alpha_k}. \tag{6}$$

Finally, we approximate $p(\theta|\alpha)$ in the simplex basis with $\hat{p}(\theta|\mu_1, \Sigma_1) = \mathcal{LN}(\theta|\mu_1, \Sigma_1)$ where $\mathcal{LN}$ is a logistic normal distribution with parameters $\mu_1, \Sigma_1$. Although we approximate the Dirichlet prior in LDA with a logistic normal, this is *not* the same idea as a correlated topic model (Blei & Lafferty, 2006), because we use a diagonal covariance matrix. Rather, it is an approximation to standard LDA.

## 3.3 Variational Objective

Now we can write the modified variational objective function. We use a logistic normal variational distribution over $\theta$ with diagonal covariance. More precisely, we define two inference networks as feed forward neural networks $f_\mu$ and $f_\Sigma$ with parameters $\boldsymbol{\delta}$; the output of each network is a vector in $\mathbb{R}^K$. Then for a document $\mathbf{w}$, we define $q(\theta)$ to be logistic normal with mean $\mu_0 = f_\mu(\mathbf{w}, \boldsymbol{\delta})$ and diagonal covariance $\Sigma_0 = \text{diag}(f_\Sigma(\mathbf{w}, \boldsymbol{\delta}))$, where diag converts a column vector to a diagonal matrix. Note that we can generate samples from $q(\theta)$ by sampling $\boldsymbol{\epsilon} \sim \mathcal{N}(0, I)$ and computing $\theta = \sigma(\boldsymbol{\mu}_0 + \boldsymbol{\Sigma}_0^{1/2} \boldsymbol{\epsilon})$.

We can now write the ELBO as

$$L(\boldsymbol{\Theta}) = \sum_{d=1}^{D} \left[ - \left( \frac{1}{2} \Big\{ tr(\boldsymbol{\Sigma}_1^{-1} \boldsymbol{\Sigma}_0) + (\boldsymbol{\mu}_1 - \boldsymbol{\mu}_0)^T \boldsymbol{\Sigma}_1^{-1} (\boldsymbol{\mu}_1 - \boldsymbol{\mu}_0) - K + \log \frac{|\boldsymbol{\Sigma}_1|}{|\boldsymbol{\Sigma}_0|} \Big\} \right) \right. \tag{7}$$

$$\left. + \mathbb{E}_{\boldsymbol{\epsilon} \sim \mathcal{N}(0,I)} \left[ \mathbf{w}_d^\top \log \left( \sigma(\boldsymbol{\beta}) \sigma(\boldsymbol{\mu}_0 + \boldsymbol{\Sigma}_0^{1/2} \boldsymbol{\epsilon}) \right) \right] \right],$$

where $\boldsymbol{\Theta}$ represents the set of all the model and variational parameters and $\mathbf{w}_1 \ldots \mathbf{w}_D$ are the documents in the corpus. The first line in this equation arises from the KL divergence between the two logistic normal distributions $q$ and $\hat{p}$, while the second line is the reconstruction error.

In order to impose the simplex constraint on the $\beta$ matrix during the optimization, we apply the softmax transformation. That is, each topic $\beta_k \in \mathbb{R}^V$ is unconstrained, and the notation $\sigma(\boldsymbol{\beta})$ means to apply the softmax function separately to each column of the matrix $\beta$. Note that the mixture of multinomials for each word $w_n$ can then be written as $p(w_n|\beta, \theta) = [\sigma(\beta)\theta]_{w_n}$, which explains the dot product in (7). To optimize (7), we use stochastic gradient descent using Monte Carlo samples from $\boldsymbol{\epsilon}$, following the Law of the Unconscious Statistician.

## 3.4 Training and Practical Considerations: Dealing with Component Collapsing

AEVB is prone to component collapsing (Dinh & Dumoulin, 2016), which is a particular type of local optimum very close to the prior belief, early on in the training. As the latent dimensionality of the model is increased, the KL regularization in the variational objective dominates, so that the outgoing decoder weights collapse for the components of the latent variable that reach close to the prior and do not show any posterior divergence. In our case, the collapsing specifically occurs because of the inclusion of the softmax transformation to produce $\theta$. The result is that the $k$ inferred topics are identical as shown in table 7.

We were able to resolve this issue by tweaking the optimization. Specifically, we train the network with the ADAM optimizer (Kingma & Ba, 2015) using high moment weight ($\beta 1$) and learning rate ($\eta$). Through training at higher rates, early peaks in the functional space can be easily avoided. The

problem is that momentum based training coupled with higher learning rate causes the optimizer to diverge. While explicit gradient clipping helps to a certain extent, we found that batch normalization (Ioffe & Szegedy, 2015) does even better by smoothing out the functional space and hence curbing sudden divergence.

Finally, we also found an increase in performance with dropout units when applied to $\theta$ to force the network to use more of its capacity.

While more prominent in the AEVB framework, the collapsing can also occurs in DMFVI if the learning offset (referred to as the $\tau$ parameter (Hofmann, 1999)) is not set properly. Interestingly, a similar learning offset or annealing based approach can also be used to down-weight the KL term in early iterations of the training to avoid local optima.

## 4 PRODLDA: LATENT DIRICHLET ALLOCATION WITH PRODUCTS OF EXPERTS

In LDA, the distribution $p(\mathbf{w}|\theta, \beta)$ is a mixture of multinomials. A problem with this assumption is that it can never make any predictions that are sharper than the components that are being mixed (Hinton & Salakhutdinov, 2009). This can result in some topics appearing that are poor quality and do not correspond well with human judgment. One way to resolve this issue is to replace this word-level mixture with a weighted product of experts which by definition is capable of making sharper predictions than any of the constituent experts (Hinton, 2002). In this section we present a novel topic model PRODLDA that replaces the mixture assumption at the word-level in LDA with a weighted product of experts, resulting in a drastic improvement in topic coherence. This is a good illustration of the benefits of a black box inference method, like AVITM, to allow exploration of new models.

### 4.1 MODEL

The PRODLDA model can be simply described as latent Dirichlet allocation where the word-level mixture over topics is carried out in natural parameter space, i.e. the topic matrix is not constrained to exist in a multinomial simplex prior to mixing. In other words, the only changes from LDA are that $\beta$ is unnormalized, and that the conditional distribution of $w_n$ is defined as $w_n|\beta, \theta \sim$ Multinomial$(1, \sigma(\beta\theta))$.

The connection to a product of experts is straightforward, as for the multinomial, a mixture of natural parameters corresponds to a weighted geometric average of the mean parameters. That is, consider two $N$ dimensional multinomials parametrized by mean vectors $\mathbf{p}$ and $\mathbf{q}$. Define the corresponding natural parameters as $\mathbf{p} = \sigma(\boldsymbol{r})$ and $\mathbf{q} = \sigma(\boldsymbol{s})$, and let $\delta \in [0, 1]$. It is then easy to show that

$$P\Big(\mathbf{x}|\delta\boldsymbol{r} + (1-\delta)\boldsymbol{s}\Big) \propto \prod_{i=1}^{N} \sigma(\delta r_i + (1-\delta)s_i)^{x_i} \propto \prod_{i=1}^{N} [r_i^{\delta} \cdot s_i^{(1-\delta)}]^{x_i}.$$

So the PRODLDA model can be simply described as a product of experts, that is, $p(w_n|\theta, \beta) \propto \prod_k p(w_n|z_n = k, \beta)^{\theta_k}$. PRODLDA is an instance of the exponential-family PCA (Collins et al., 2001) class, and relates to the exponential-family harmoniums (Welling et al., 2004) but with non-Gaussian priors.

## 5 RELATED WORK

For an overview of topic modeling, see Blei (2012). There are several examples of topic models based on neural networks and neural variational inference (Hinton & Salakhutdinov, 2009; Larochelle & Lauly, 2012; Mnih & Gregor, 2014; Miao et al., 2016) but we are unaware of methods that apply AEVB generically to a topic model specified by an analyst, or even of a successful application of AEVB to the most widely used topic model, latent Dirichlet allocation.

Recently, Miao et al. (2016) introduced a closely related model called the Neural Variational Document Model (NVDM). This method uses a latent Gaussian distribution over topics, like probabilistic latent semantic indexing, and averages over topic-word distributions in the logit space. However,

they do not use either of the two key aspects of our work: explicitly approximating the Dirichlet prior using a Gaussian, or high-momentum training. In the experiments we show that these aspects lead to much improved training and much better topics.

# 6    EXPERIMENTS AND RESULTS

Qualitative evaluation of topic models is a challenging task and consequently a large body of work has developed automatic evaluation metrics that attempt to match human judgment of topic quality. Traditionally, perplexity has been used to measure the goodness-of-fit of the model but it has been repeatedly shown that perplexity is not a good metric for qualitative evaluation of topics (Newman et al., 2010). Several new metrics of topic coherence evaluation have thus been proposed; see Lau et al. (2014) for a comparative review. Lau et al. (2014) showed that among all the competing metrics, normalized pointwise mutual information (NPMI) between all the pairs of words in a set of topics matches human judgment most closely, so we adopt it in this work. We also report perplexity, primarily as a way of evaluating the capability of different optimizers. Following standard practice (Blei et al., 2003), for variational methods we use the ELBO to calculate perplexity. For AEVB methods, we calculate the ELBO using the same Monte Carlo approximation as for training.

We run experiments on both the *20 Newsgroups* (11,000 training instances with 2000 word vocabulary) and *RCV1 Volume 2* ( 800K training instances with 10000 word vocabulary) datasets. Our preprocessing involves tokenization, removal of some non UTF-8 characters for 20 Newsgroups and English stop word removal. We first compare our AVITM inference method with the standard online mean-field variational inference (Hoffman et al., 2010) and collapsed Gibbs sampling (Griffiths & Steyvers, 2004) on the LDA model. We use standard implementations of both methods, `scikit-learn` for DMFVI and `mallet` (McCallum, 2002) for collapsed Gibbs. Then we compare two autoencoding inference methods on three different topic models: standard LDA, PRODLDA using our inference method and the Neural Variational Document Model (NVDM) (Miao et al., 2016), using the inference described in the paper.[2]

| # topics | ProdLDA VAE | LDA VAE | LDA DMFVI | LDA Collapsed Gibbs | NVDM |
|---|---|---|---|---|---|
| 50 | **0.24** | 0.11 | 0.11 | 0.17 | 0.08 |
| 200 | **0.19** | 0.11 | 0.06 | 0.14 | 0.06 |

Table 1: Average topic coherence on the 20 Newsgroups dataset. Higher is better.

Tables 1 and 2 show the average topic coherence values for all the models for two different settings of $k$, the number of topics. Comparing the different inference methods for LDA, we find that, consistent with previous work, collapsed Gibbs sampling yields better topics than mean-field methods. Among the variational methods, we find that VAE-LDA model (AVITM) [3] yields similar topic coherence and perplexity to the standard DMFVI (although in some cases, VAE-LDA yields significantly better topics). However, AVITM is significantly faster to train than DMFVI. It takes 46 seconds on 20 Newsgroup compared to 18 minutes for DMFVI. Whereas for a million document corpus of RCV1 it only under 1.5 hours while *scikit-learn's* implementation of DMFVI failed to return any results even after running for 24 hours.[4]

Comparing the new topic models than LDA, it is clear that PRODLDA finds significantly better topics than LDA, even when trained by collapsed Gibbs sampling. To verify this qualitatively, we display examples of topics from all the models in Table 6. The topics from ProdLDA appear visually more coherent than NVDM or LDA. Unfortunately, NVDM does not perform comparatively to LDA

---

[2]We have used both `https://github.com/carpedm20/variational-text-tensorflow` and the NVDM author's (Miao et al., 2016) implementation.

[3]We recently found that 'whitening' the topic matrix significantly improves the topic coherence for VAE-LDA. Manuscript in preparation.

[4]Therefore, we were not able to report topic coherence for DMFVI on RCV1

| # topics | ProdLDA VAE | LDA VAE | LDA DMFVI | LDA Collapsed Gibbs | NVDM |
|---|---|---|---|---|---|
| 50 | **0.14** | 0.07 | - | 0.04 | 0.07 |
| 200 | **0.12** | 0.05 | - | 0.06 | 0.05 |

Table 2: Average topic coherence on the RCV1 dataset. Higher is better. Results not reported for LDA DMFVI, as inference failed to converge in 24 hours.

| # topics | ProdLDA VAE | LDA VAE | LDA DMFVI | LDA Collapsed Gibbs | NVDM |
|---|---|---|---|---|---|
| 50 | 1172 | 1059 | 1046 | **728** | 837 |
| 200 | 1168 | 1128 | 1195 | **688** | 884 |

Table 3: Perplexity scores for 20 Newsgroups. Lower is better.

for any value of $k$. To avoid any training dissimilarities we train all the competing models until we reach the perplexities that were reported in previous work. These are reported in Table 3[5].

A major benefit of AVITM inference is that it does not require running variational optimization, which can be costly, for new data. Rather, the inference network can be used to obtain topic proportions for new documents for new data points without running any optimization. We evaluate whether this approximation is accurate, i.e. whether the neural network effectively learns to mimic probabilistic inference. We verify this by training the model on the training set, then on the test set, holding the topics ($\beta$ matrix) fixed, and comparing the test perplexity if we obtain topic proportions by running the inference neural network directly, or by the standard method of variational optimization of the inference network on the test set. As shown in Table 4, the perplexity remains practically un-changed. The computational benefits of this are remarkable. On both the datasets, computing perplexity using the neural network takes well under a minute, while running the standard variational approximation takes $\sim 3$ minutes even on the smaller 20 Newsgroups data. Finally, we investigate the reasons behind the improved topic coherence in PRODLDA. First, Table 5 explores the effects of each of our two main ideas separately. In this table, "Dirichlet" means that the prior is the Laplace approximation to Dirichlet($\alpha = 0.02$), while "Gaussian" indicates that we use a standard Gaussian as prior. 'High Learning Rate' training means we use $\beta1 > 0.8$ and $0.1 > \eta > 0.001$[6] with batch normalization, whereas "Low Learning Rate" means $\beta1 > 0.8$ and $0.0009 > \eta > 0.00009$ without batch normalization. (For both parameters, the precise value was chosen by Bayesian optimization. We found that these values in the "*with BN*" cases were close to the default settings in the Adam optimizer.) We find that the high topic coherence that we achieve in this work is only possible if we use both tricks together. In fact the high learning rates with momentum is required to avoid local minima in the beginning of the training and batch-normalization is required to be able to train the network at these values without diverging. If trained at a lower momentum value or at a lower learning rate PRODLDA shows component collapsing. Interestingly, if we choose a Gaussian prior, rather than the logistic normal approximation used in ProdLDA or NVLDA, the model is easier to train even with low learning rate without any momentum or batch normalization.

The main advantage of AVITM topic models as opposed to NVDM is that the Laplace approximation allows us to match a specific Dirichlet prior of interest. As pointed out by Wallach et al. (2009), the choice of Dirichlet hyperparameter is important to the topic quality of LDA. Following this reasoning, we hypothesize that AVITM topics are higher quality than those of NVDM because they are much more focused, i.e., apply to a more specific subset of documents of interest. We provide support for this hypothesis in Figure 1, by evaluating the sparsity of the posterior proportions over topics, that is, how many of the model's topics are typically used to explain each document. In order to estimate the sparsity in topic proportions, we project samples from the Gaussian latent spaces of PRODLDA and NVDM in the simplex and average them across documents. We compare the topic

---

[5]We note that much recent work follows Hinton & Salakhutdinov (2009) in reporting perplexity for the LDA Gibbs sampler on only a small subset of the test data. Our results are different because we use the entire test dataset.

[6]$\beta1$ is the weight on the average of the gradients from the previous time step and $\eta$ refers to the learning rate.

| # Topics | Inference Network Only | Inference Network + Optimization |
|---|---|---|
| **50** | 1172 | 1162 |
| **200** | 1168 | 1151 |

Table 4: Evaluation of inference network of VAE-LDA on 20 Newsgroups test set. "Inference network only" shows the test perplexity when the inference network is trained on the training set, but no variational optimization is performed on the test set. "Inference Network + Optimization" shows the standard approach of optimizing the ELBO on the test set. The neural network effectively learns to approximate probabilistic inference effectively.

sparsity for the standard Gaussian prior used by NVDM to the Laplace approximation of Dirichlet priors with different hyperparameters. Clearly the Laplace approximation to the Dirichlet prior significantly promotes sparsity, providing support for our hypothesis that preserving the Dirichlet prior explains the the increased topic coherence in our method.

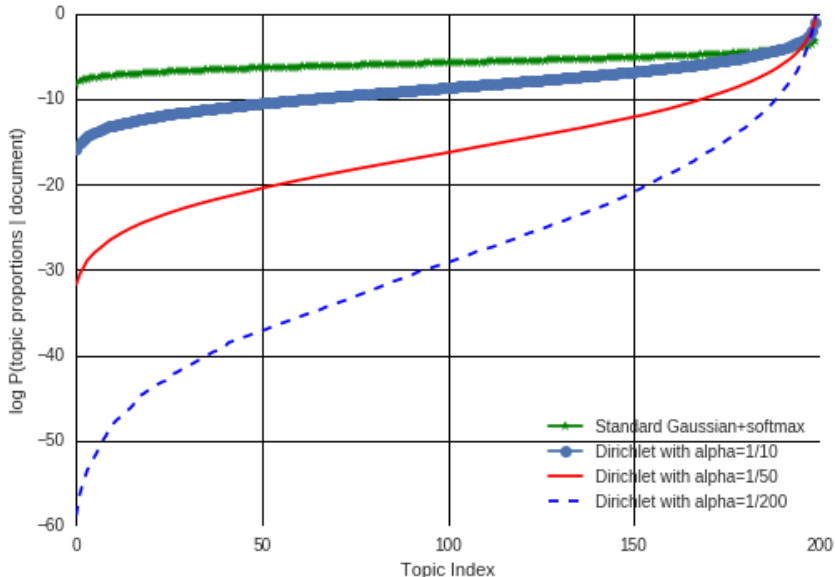

Figure 1: Effect of prior assumptions on $\theta$ on sparsity of $\theta$ in neural topic models.

| | Dirichlet +High Learning Rate | Dirichlet +Low Learning Rate | Gaussian Prior +High Learning Rate | Gaussian Prior +Low Learning Rate |
|---|---|---|---|---|
| Topic Coherence | **0.24** | 0.016 | 0.08 | 0.08 |

Table 5: Average topic coherence for different choices of prior and optimization strategies of PRODLDA on 20 Newsgroup for $k = 50$.

The inference network architecture can be found in figure 2 in the appendix.

## 7 DISCUSSION AND FUTURE WORK

We present what is to our knowledge the first effective AEVB inference algorithm for latent Dirichlet allocation. Although this combination may seem simple in principle, in practice this method is difficult to train because of the Dirichlet prior and because of the component collapsing problem. By addressing both of these problems, we presented a black-box inference method for topic models with the notable advantage that the neural network allows computing topic proportions for new documents without the need to run any variational optimization. As an illustration of the advantages of

| Model | Topics |
|---|---|
| **ProdLDA** | motherboard meg printer quadra hd windows processor vga mhz connector<br>armenian genocide turks turkish muslim massacre turkey armenians armenia greek<br>voltage nec outlet circuit cable wiring wire panel motor install<br>season nhl team hockey playoff puck league flyers defensive player<br>israel israeli lebanese arab lebanon arabs civilian territory palestinian militia |
| **LDA<br>NVLDA** | db file output program line entry write bit int return<br>drive disk get card scsi use hard ide controller one<br>game team play win year player get think good make<br>use law state health file gun public issue control firearm<br>people say one think life make know god man see |
| **LDA<br>DMFVI** | write article dod ride right go get night dealer like<br>gun law use drug crime government court criminal firearm control<br>lunar flyers hitter spacecraft power us existence god go mean<br>stephanopoulos encrypt spacecraft ripem rsa cipher saturn violate lunar crypto<br>file program available server version include software entry ftp use |
| **LDA<br>Collapsed Gibbs** | get right back light side like see take time one<br>list mail send post anonymous internet file information user message<br>thanks please know anyone help look appreciate get need email<br>jesus church god law say christian one christ day come<br>bike dod ride dog motorcycle write article bmw helmet get |
| **NVDM** | light die burn body life inside mother tear kill christian<br>insurance drug different sport friend bank owner vancouver buy prayer<br>input package interface output tape offer component channel level model<br>price quadra hockey slot san playoff jose deal market dealer<br>christian church gateway catholic christianity homosexual resurrection modem mouse sunday |

Table 6: Five *randomly* selected topics from all the models.

> 1. write article get thanks like anyone please know look one
> 2. article write one please like anyone know make want get
> 3. write article thanks anyone please like get one think look
> 4. article write one get like know thanks anyone try need
> 5. article write thanks please get like anyone one time make

Table 7: VAE-LDA fails to learn any meaningful topics when component collapsing occurs. The table shows five randomly sampled topics (, which are essentially slight variants of each other) from when the VAE-LDA model is trained without BN and high momentum training.

black box inference techniques, we presented a new topic model, ProdLDA, which achieves significantly better topics than LDA, while requiring a change of only one line of code from AVITM for LDA. Our results suggest that AVITM inference is ready to take its place alongside mean field and collapsed Gibbs as one of the workhorse inference methods for topic models. Future work could include extending our inference methods to handle dynamic and correlated topic models.

ACKNOWLEDGMENTS

We thank Andriy Mnih, Chris Dyer, Chris Russell, David Blei, Hannah Wallach, Max Welling, Mirella Lapata and Yishu Miao for helpful comments, discussions and feedback.

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

## A  NETWORK ARCHITECTURE

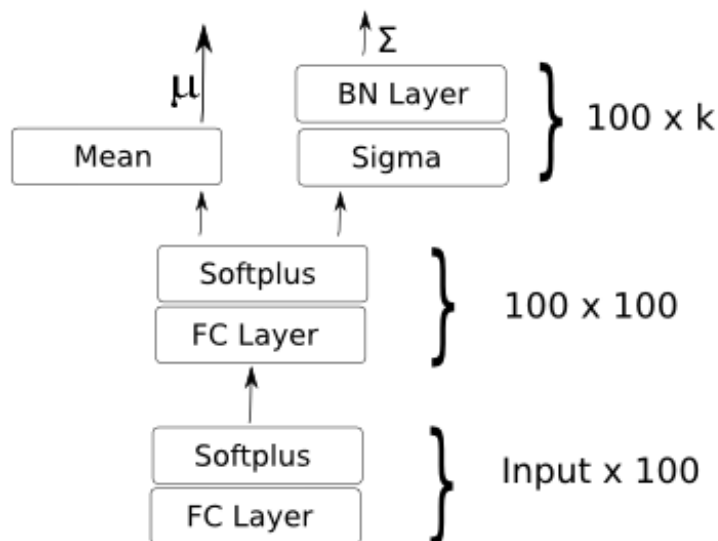

Figure 2: Architecture of the inference network used in the experiments.

