# Peer review of "Autoencoding Variational Inference For Topic Models"

_ICLR 2017 — accepted_

[Public Comment · Erik Holmer · 08 Nov 2016]
**Perplexity**

The perplexity you're reporting for the 20 Newsgroups dataset using LDA Collapsed Gibbs is better than for any other method I've seen.  Would you mind sharing the parameters you used and/or the preprocessed dataset?

[Public Comment · (anonymous) · rating 5 · confidence 4 · 05 Dec 2016]
**Comparison to NVDM looks unfair**

The comparison to NVDM looks unfair since the user introduces a couples tricks (Dirichlet prior, batch normalisation, high momentum training, etc.) which NVDM doesn't use. A more convincing experimental design is to explore the effect of each trick separately in neural variational inference.

[Official Review · AnonReviewer3 · rating 6 · confidence 5 · 14 Dec 2016]
**VAE model for LDA. Interesting idea, but a incremental.**

This is an interesting paper on a VAE framework for topic models. The main idea is to train a recognition model for the inference phase which, because of so called “amortized inference” can be much faster than normal inference where inference must be run iteratively for every document. Some comments:
Eqn 5: I find the notation p(theta(h)|alpha) awkward. Why not P(h|alpha) ?
The generative model seems agnostic to document length, meaning that the latent variables only generate probabilities over word space. However, the recognition model is happy to radically change the probabilities q(z|x) if the document length changes because the input to q changes. This seems undesirable. Maybe they should normalize the input to the recognition network?
The ProdLDA model might well be equivalent to exponential family PCA or some variant thereof:

[Official Review · AnonReviewer4 · rating 7 · confidence 3 · 16 Dec 2016 (modified: 18 Dec 2016)]
**Nice paper to read**

This paper proposes the use of neural variational inference method for topic models. The paper shows a nice trick to approximate Dirichlet prior using softmax basis with a Gaussian and then the model is trained to maximize the variational lower bound. Also, the authors study a better way to alleviate the component collapsing issue, which has been problematic for continuous latent variables that follow Gaussian distribution. The results look promising and the experimental protocol sounds fine.

Minor comments:
Please add citation to [1] or [2] for neural variational inference, and [2] for VAE. 
A typo in “This approximation to the Dirichlet prior p(θ|α) is results in the distribution”, it should be “This approximation to the Dirichlet prior p(θ|α) results in the distribution”

In table 2, it is written that DMFVI was trained more than 24hrs but failed to deliver any result, but why not wait until the end and report the numbers?

In table 3, why are the perplexities of LDA-Collapsed Gibbs and NVDM are lower while the proposed models (ProdLDA) generates more coherent topics? What is your intuition on this?

How does the training speed (until the convergence) differs by using different learning-rate and momentum scheduling approaches shown as in figure 1?

It may be also interesting to add some more analysis on the latent variables z (component collapsing and etc., although your results indirectly show that the learning-rate and momentum scheduling trick removes this issue).

Overall, the paper clearly proposes its main idea, explain why it is good to use NVI, and its experimental results support the original claim. It explains well what are the challenges and demonstrate their solutions. 

[1] Minh et al., Neural Variational Inference and Learning in Belief Networks, ICML’14
[2] Rezende et al., Stochastic Backpropagation and Approximate Inference in Deep Generative Models, ICML’14

[Official Review · AnonReviewer1 · rating 6 · confidence 4 · 17 Dec 2016 (modified: 21 Jan 2017)]
**Promising direction, but the paper needs more work**

The authors propose NVI for LDA variants. The authors compare NVI-LDA to standard inference schemes such as CGS and online SVI. The authors also evaluate NVI on a different model ProdLDA (not sure this model has been proposed before in the topic modeling literature though?)

In general, I like the direction of this paper and NVI looks promising for LDA. The experimental results however confound model vs inference which makes it hard to understand the significance of the results. Furthermore, the authors don't discuss hyper-parameter selection which is known to significantly impact performance of topic models. This makes it hard to understand when the proposed method can be expected to work. 

Can you maybe generate synthetic datasets with different Dirichlet distributions and assess when the proposed method recovers the true parameters?

Figure 1: Is this prior or posterior? The text talks about sparsity whereas the y-axis reads "log p(topic proportions)" which is a bit confusing.

Section 3.2: it is not clear what you mean by unimodal in softmax basis. Consider a Dirichlet on K-dimensional simplex with concentration parameter alpha/K where alpha<1 makes it multimodal. Isn't the softmax basis still multimodal?

None of the numbers include error bars. Are the results statistically significant?


Minor comments:

Last term in equation (3) is not "error"; reconstruction accuracy or negative reconstruction error perhaps?

The idea of using an inference network is much older, cf. Helmholtz machine.

[Final Decision · Program Chairs · 06 Feb 2017]
**ICLR committee final decision**

The reviewers agree that the approach is interesting and the paper presents useful findings. They also raise enough questions and suggestions for improvements that I believe the paper will be much stronger after further revision, though these seem straightforward to address.